# Allopurinol to reduce cardiovascular morbidity and mortality: A systematic review and meta-analysis

Karel H. van der Pol[1], Kimberley E. Wever[2], Mariette Verbakel[3], Frank L. J. Visseren[4], Jan H. Cornel[5,6], Gerard A. Rongen[1,7]*

**1** Department of Pharmacology and Toxicology, Radboud Institute for Health Sciences, Radboudumc, Nijmegen, The Netherlands, **2** Department for Health Evidence, Radboud Institute for Health Sciences, Radboudumc, Nijmegen, The Netherlands, **3** Harteraad, Nijmegen, The Netherlands, **4** Department of Vascular Medicine, UMC Utrecht, Utrecht, The Netherlands, **5** Department of Cardiology, Radboud Institute for Health Sciences, Radboudumc, Nijmegen, The Netherlands, **6** Department of Cardiology, Northwest Clinics, Alkmaar, The Netherlands, **7** Department of Internal Medicine, Radboud Institute for Health Sciences, Radboudumc, Nijmegen, The Netherlands

* Gerard.Rongen@Radboudumc.nl

**Data Availability Statement:** All relevant data are within the manuscript and its Supporting Information files.

**Funding:** The author(s) received no specific funding for this work.

## Abstract

### Aims

To compare the effectiveness of allopurinol with no treatment or placebo for the prevention of cardiovascular events in hyperuricemic patients.

### Methods and results

Pubmed, Web of Science and Cochrane library were searched from inception until July 2020. Randomized controlled trials (RCT) and observational studies in hyperuricemic patients without significant renal disease and treated with allopurinol, versus placebo or no treatment were included. Outcome measures were cardiovascular mortality, myocardial infarction, stroke, or a combined endpoint (CM/MI/S). For RCT's a random effects meta-analysis was performed. For observational studies a narrative synthesis was performed. Of the original 1995 references we ultimately included 26 RCT's and 21 observational studies. We found a significantly reduced risk of combined endpoint (Risk Ratio 0.65 [95% CI] [0.46 to 0.91]; p = 0.012) and myocardial infarction (RR 0.47 [0.27 to 0.80]; p = 0.01) in the allopurinol group compared to controls. We found no significant effect of allopurinol on stroke or cardiovascular mortality. Of the 15 observational studies with sufficient quality, allopurinol was associated with reduced cardiovascular mortality in 1 out of 3 studies that reported this outcome, myocardial infarction in 6 out of 8, stroke in 4 out of 7, and combined end-point in 2 out of 2. Cardiovascular benefit was only observed when allopurinol therapy was prolonged for more than 6 months and when an appropriate allopurinol dose was administered (300 mg or more/day) or sufficient reduction of serum urate concentration was achieved (<0.36 mmol/l).

**Competing interests:** Dr. Cornel reports grants from ZonMw (LoDoCo 2 trial), personal fees from Amgen (advisory board), personal fees from Servier (advisory board), personal fees from Astra Zeneca (advisory board), outside the submitted work; All other authors declare: no support from any organisation for the submitted work; no financial relationships with any organisations that might have an interest in the submitted work in the previous three years; no other relationships or activities that could appear to have influenced the submitted work. This does not alter our adherence to PLOS ONE policies on sharing data and materials.

## Conclusions

Data from RCT's and observational studies indicate that allopurinol treatment reduces cardiovascular risk in patients with hyperuricemia. However, the quality of evidence from RCTs is low to moderate. To establish whether allopurinol lowers the risk of cardiovascular events a well-designed and adequately powered randomized, placebo-controlled trial is needed in high-risk patients with hyperuricemia.

## Systematic review registration

PROSPERO registration CRD42018089744

## Introduction

Allopurinol, a well-known xanthine oxidase inhibitor, has been used for over 50 years to lower serum uric acid. According to clinical guidelines, this drug should be used to treat complications that result from hyperuricemia: frequent gout attacks, urate arthropathy, urate depositions in the skin (tophi) and urate nephropathy. Although urate nephropathy is not a well-defined entity, prove of urate stones and prevention of renal decline that results from tumour lysis syndrome are accepted conditions to start allopurinol [1, 2]. Considerably less agreement exists in clinical practice on the use of allopurinol in patients with asymptomatic or minimally symptomatic hyperuricemia, e.g. in patients who only rarely suffer a gout attack that can easily be managed by colchicine, prednisone or a cyclooxygenase(COX)-inhibitor. Hyperuricemia is associated with cardiovascular morbidity and mortality [3, 4], and an increased risk of renal failure [5]. Clinical guidelines on the use of a xanthine oxidase inhibitor differ in particular for patients with elevated cardiovascular risk or renal insufficiency and otherwise asymptomatic hyperuricemia. For example, Dutch guidelines for primary physicians do not recommend allopurinol for these patients [6, 7]. In contrast, the guidelines for rheumatologists suggest to discuss the option of allopurinol treatment in patients with cardiovascular disease or multiple cardiovascular risk factors, but do not state the arguments pro- or contra- that could be used in making this shared decision with the patient [7]. Three recently published randomized controlled trials (RCTs) fuelled this discussion further: the Cardiovascular Safety of Febuxostat and Allopurinol in Patients with Gout and Cardiovascular Morbidities trial (CARES) [8], the Febuxostat for Cerebral and Cardiorenovascular Events Prevention Study (FREED) [9], and the Long-term Cardiovascular Safety of Febuxostat Compared with Allopurinol in Patients with Gout Trial (FAST) [10]. The CARES trial was a large (>4000 patients with hyperuricemia and gout) randomized double blind clinical trial that compared febuxostat with allopurinol with cardiovascular safety as primary outcome. The patients who had been allocated to febuxostat showed more reduction in serum urate. However cardiovascular mortality was significantly lower in the group allocated to allopurinol as compared with febuxostat [8]. This finding has subsequently been substantiated by real life observational data in large cohorts [11, 12]. The FAST trial was a large (>6000 patients with gout and at least one additional cardiovascular risk factor) randomized, open-label, blinded-endpoint, non-inferiority trial of febuxostat versus allopurinol again with cardiovascular safety as the primary outcome. In this trial febuxostat was non-inferior to allopurinol for cardiovascular mortality, myocardial infarction, stroke or combined outcome [10], contrasting with the findings of the previously published CARES trial. This might be explained by a different incidence of cardiovascular disease in the

medical history of the included patients. All patients included in the CARES trial had a history of previous cardiovascular disease as compared with only thirty-three percent of patients in the FAST trial. It should also be noted that rate of treatment discontinuation (25.3% vs 56.6%) and loss to follow-up (5.8% vs 45%) was lower in the FAST trial than the CARES trial. The FREED trial was a randomized trial with PROBE design comparing febuxostat with non-febuxostat therapy in around 1200 patients (age > 65 years) with asymptomatic hyperuricemia. Cardiovascular events, mortality and renal function were the composite primary outcome. Although febuxostat did not affect cardiovascular events in this trial, the decline in renal function was inhibited in those allocated to febuxostat as compared with usual therapy. Unfortunately, sample size was relatively low and 27.2% of the patients allocated to non-febuxostat arm used 100 mg allopurinol/day complicating the interpretation of the results.

This paucity of data on efficacy of allopurinol to prevent cardiovascular events in patients with asymptomatic or mildly symptomatic hyperuricemia triggered us to systematically review the literature on the efficacy of allopurinol compared with no treatment or placebo to prevent cardiovascular events in patients with hyperuricemia and normal or moderately reduced renal function.

## Methods

This review is reported according to the PRISMA guidelines. The full systematic review protocol was prospectively submitted at the PROSPERO international prospective register of systematic reviews in March 2018 (registration number CRD42018089744, available from https://www.crd.york.ac.uk/prospero/display_record.php?ID=CRD42018089744). All amendments to the review protocol were also registered on PROSPERO.

### Search and study selection process

The comprehensive search string is included as S1 File. On July 22 2020, we searched PubMed, Web of Science and Cochrane library for RCTs or observational (cohort) studies in human adults with hyperuricemia without severe renal disease (defined as Modification of Diet in Renal Disease (MDRD) formula < 30 ml/min/1.73 m$^2$) who were treated with a xanthine oxidase inhibitor (allopurinol or febuxostat) in any dose regimen or treatment duration. No restrictions on publication date or language were applied. Automated duplicate removal was used to remove duplicates (i.e. studies occurring more than once in our database after the search), if a DOI occurred more than once in the database the duplicate entries were automatically removed.

Two reviewers (KvdP, KW or GR) independently performed screening for eligibility based on title and abstract and assessment for final inclusion based on full-text. In case of discrepancies, the reviewers reached consensus through discussion. For details on the exclusion criteria see the flow chart of the study selection process (Fig 1) and the PROSPERO protocol. Only studies that used appropriate controls (placebo, no treatment) were included.

### Data extraction and risk of bias or quality assessment

Extraction of study characteristics and outcome data was performed by KvdP and independently checked by two reviewers (GR and JHC). Outcome data were extracted for four predefined primary outcomes: incidence of cardiovascular mortality, incidence of stroke, incidence of myocardial infarction and the combined incidence of these three. The risk of bias in RCTs was assessed by two independent using the Cochrane risk of bias tool in RevMan 5.3 [13]. The New-Castle Ottowa scale was used to assess the quality of observational studies (available at http://www.ohri.ca/programs/clinical_epidemiology/oxford.asp). For randomized cross-over

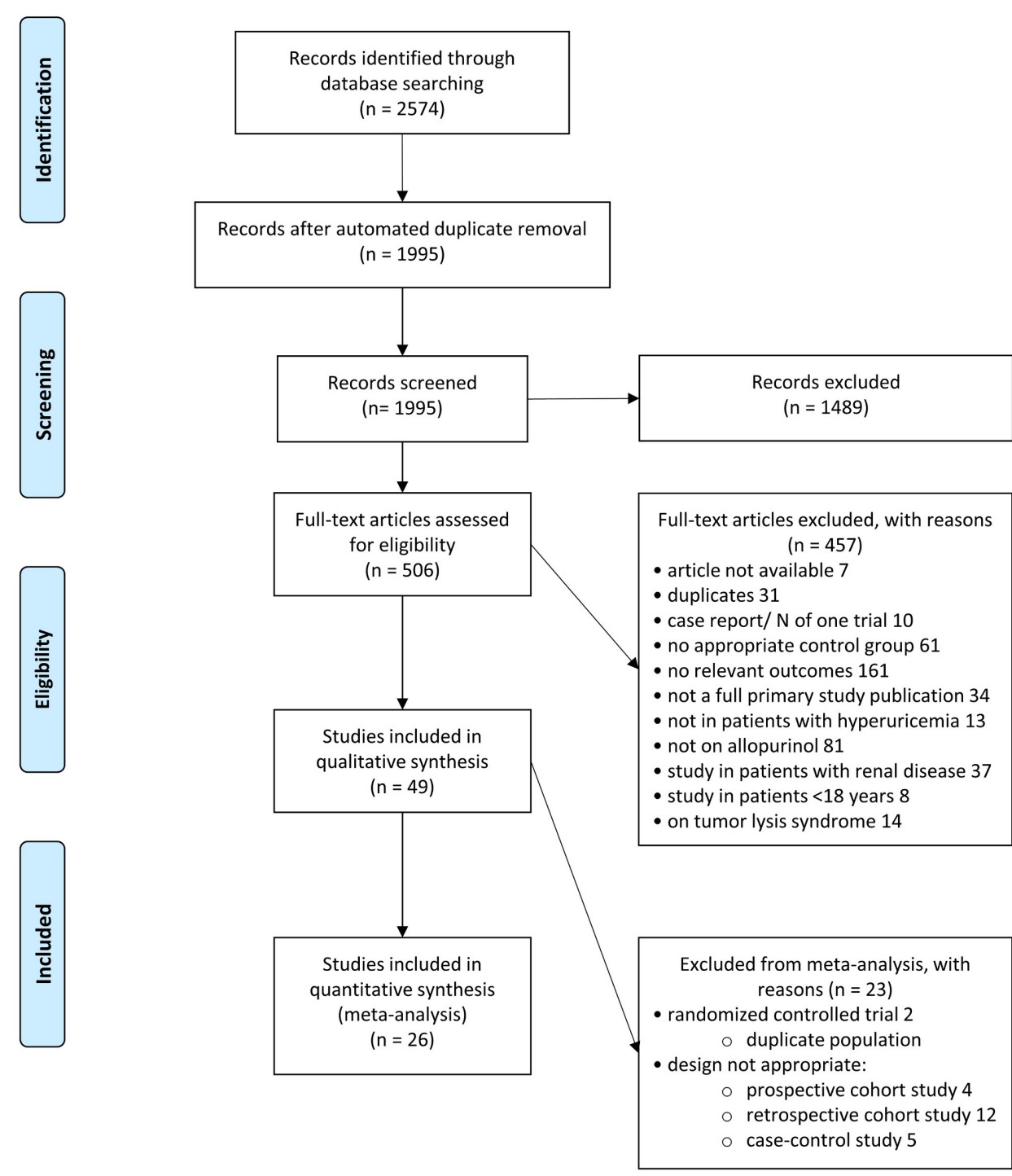

**Fig 1. Flow diagram of study selection process.** *there were 4 publications of 2 RCTs. Only 2 of these publications were included in the meta-analysis the other 2 were excluded.

studies, we assessed risk of bias using the systems provided in the Cochrane handbook. All assessments were performed by independent reviewers (KvdP, GR or KW). In case of discrepancies, the reviewers reached consensus through discussion.

## Data synthesis strategy

Each of the four outcomes was reported in at least 3 of the retrieved RCTs. In accordance with our predefined data-analysis plan, a meta-analysis was performed on each of these outcomes. The meta-analysis was performed using the OpenMeta[analyst] software package [14]. We calculated the risk ratio and corresponding 95% confidence interval (RR [95%CI]) and individual study effects were pooled using the DerSimonian and Laird random effects model. To include studies in the meta-analysis in which zero events occurred the numerator 0 was substituted with 0.5 (i.e. correction factor). Heterogeneity was assessed using the $I^2$, where an $I^2$ of >50% was considered substantial. In the predefined protocol subgroup analysis was only to be used to explore causes of heterogeneity in the results. Planned subgroup analyses on sex, age, dose, and various comorbidities were not performed because heterogeneity was 0% in all four overall analyses. For the same reason, reporting $R^2$ was redundant and therefore omitted. Publication bias was assessed if at least 10 studies reported a certain outcome. We used visual inspection, Egger's regression and trim and fill analysis to test for funnel plot asymmetry. For observational studies we performed a narrative synthesis for the four primary outcomes, taking into account the type of intervention and treatment regime and characteristics of the study population. A meta-analysis of observational studies was not included in the original protocol and therefore not performed. It was expected that there would be significant differences in design of observational studies resulting in large methodological and clinical heterogeneity. The Grading of Recommendations Assessment, Development and Evaluation (GRADE) criteria were used to assess the overall quality of the body of evidence [15–17]. A summary of findings table was created using GRADE PRO [18].

## Patient and public involvement

Patients were involved in writing the discussion section dealing with communication of results to patients, in order to support shared decision making on the initiation of allopurinol in patients with asymptomatic or mildly symptomatic hyperuricemia. No patients were involved in designing or executing the study.

# Results

## Search and study selection

A flow chart of the study selection process is shown in Fig 1. After automated duplicate removal, 1995 unique publications were retrieved from Pubmed and Web of Science. The Cochrane library search did not reveal additional unique publications. After title and abstract screening, 506 eligible publications remained, which were assessed for final inclusion based on full-text. Ultimately, 49 publications met our inclusion criteria and underwent data extraction and quality assessment (26 publications of RCTs, 2 cross-over studies and 21 observational studies). In almost all trials cardiovascular events were not the primary outcome. We therefore extracted data on cardiovascular events from the reported adverse events in those trials. Therefore, trials not reporting adverse events were excluded.

## Study characteristics

A complete list of all included studies and their general characteristics (e.g. sex and age of the participants, duration and dose of treatment) is presented as S1 Table.

Twenty-six publications of 24 RCTs reporting cardiovascular outcomes were retrieved [19–44]. One trial was published twice: once after two years of follow-up [22], and once after an additional 5 years follow up after patients had returned to pre-trial medication [43]. The latter

publication contained more details about cardiovascular events and was therefore used in the meta-analysis. Two other publications appear to be on the same study population, with one reporting on carotid intima-media thickness [28], and the other on renal function [27]. The two publications report identical data on cardiovascular events, which were included once in our meta-analysis. Two randomized crossover trials were included [45, 46]. In the included trials, a total of 3080 patients have been allocated to either allopurinol (n = 1638) or no-urate-lowering therapy (n = 1442). In total 21 observational studies were included: 16 cohort studies (4 prospective [47–50] and 12 retrospective [51–62]), and 5 case-control studies [63–67].

## Risk of bias and study quality

The risk of bias did not differ between outcome measures, therefore the results presented apply to all four outcomes. The individual risk of bias assessments of the 26 publications of 24 RCTs is shown as S1 Fig and S2 Table (for two randomized cross-over studies). For RCTs, a summary of the risk of bias assessment is presented in Fig 2. Out of all 26 trials, one trial was at low risk of all types of bias. Eight trials were at unclear risk of at least one type of bias. Seventeen trials were at high risk for at least one type of bias, often due to lack of blinding of selection, performance, or outcome detection. Overall, the quality of evidence derived from the RCT was classified as low for all outcomes. The individual quality assessments of the 21 observational studies are presented as S2 Table. All 5 case control studies were assessed as being of good quality. Six cohort studies were assessed as being poor quality [47–50, 54, 58], and these are therefore not included in the main text of this review (a complete overview of all studies, including those of poor quality, is available as S1 Table). The 10 remaining cohort studies were all assessed as being of good quality.

## Data synthesis—randomized clinical trials

In general, few of the included trials reported events of cardiovascular mortality, myocardial infarction, stroke, or the combined incidence of these three. For the combined outcome 6 out of 26 trials reported an event, with 39 events in 1550 patients in the allopurinol treated group and 64 events in 1354 patients in the control arm resulting in a relative risk of 0.65 (95% CI

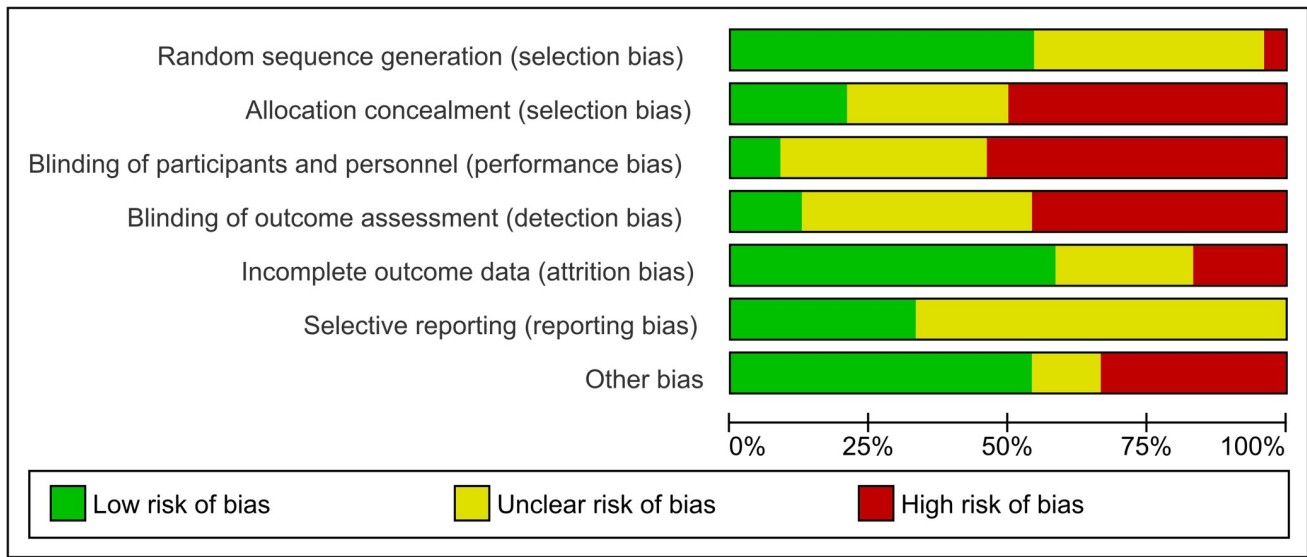

**Fig 2. Overview of risk of bias in included RCTs.**

0.46 to 0.91; p = 0.012, $I^2$ = 0%) in favour of allopurinol (Fig 3, for weights assigned to individual trials included in the meta-analysis see S3 Table). Allopurinol did not significantly affect cardiovascular mortality or stroke but led to a significant reduction in myocardial infarction (S2 Fig). Twenty out of 26 trials (including 2 cross-over studies) reported zero events. In these 20 trials 1059 patients were treated with allopurinol and 907 patients received placebo. Thus, studies that reported events included only a minority of randomized patients (938 out of the 2904 randomized patients).

## Publication bias assessment

Funnel plots for all four outcomes are shown in Figs 4 and S3. In all funnel plots a cluster of datapoints around the risk ratio of 1 and a standard error of 2 is observed, due to the large number of trials reporting 0 events in both treatment groups leads. Egger's regression test was not significant for cardiovascular mortality and stroke (respectively p = 0.49 and p = 0.14) but indicated the presence of small study effects for myocardial infarction and the combined outcome (respectively p<0.0001 and p = 0.039). Trim and fill analysis indicated respectively 0, 13, 12 ad 9 missing studies for cardiovascular mortality, myocardial infarction, stroke, and the combined outcome. However, the aforementioned lack of diversity in the effect size and standard error of the datapoints hampers funnel plot asymmetry testing, which reduces the reliability of these results.

## Data synthesis—observational studies

Table 1 summarizes the findings of retrospective cohort studies and case control studies on cardiovascular mortality, myocardial infarction, stroke, and the combined outcome of cardiovascular mortality, myocardial infarction, and stroke.

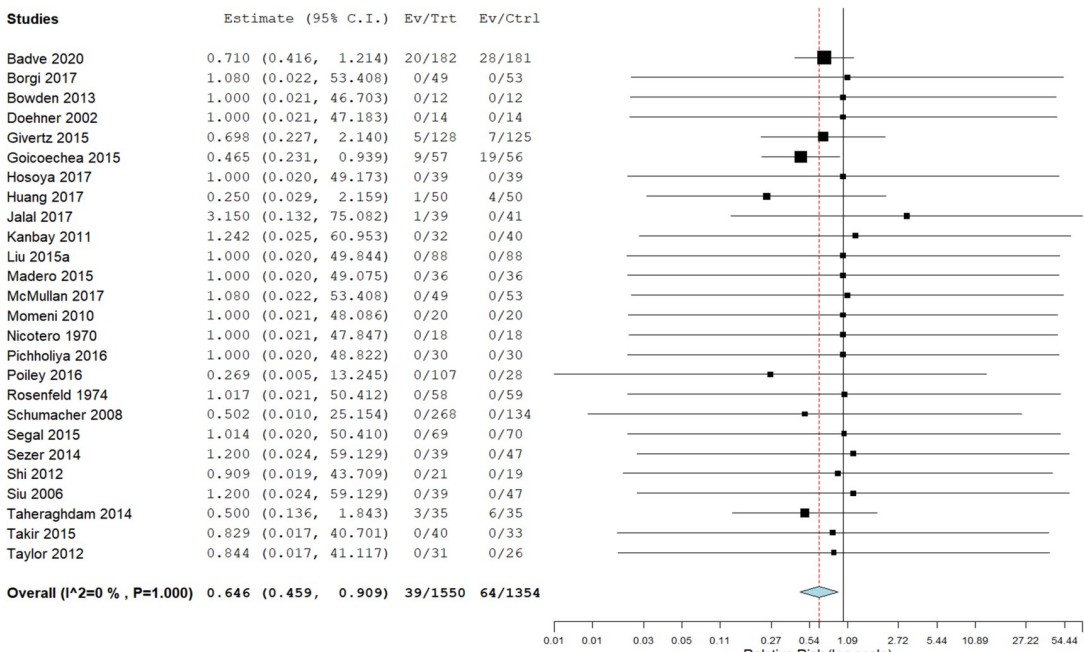

**Fig 3. Forest plot of combined outcome (cardiovascular mortality, myocardial infarction and stroke).** Overall effect of allopurinol on the combined outcome: p = 0.01.

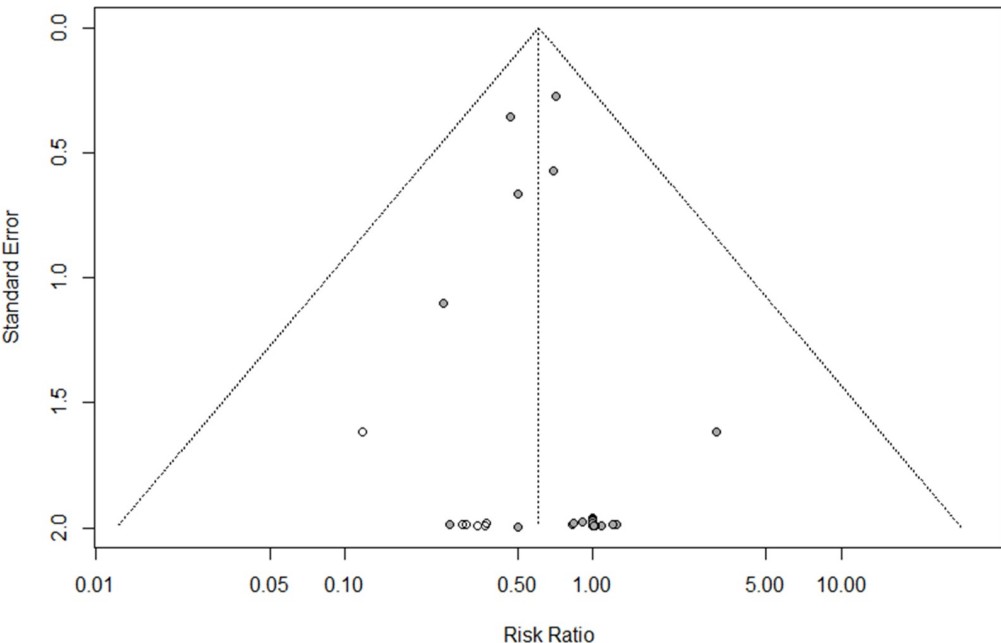

**Fig 4. Funnel plot of combined outcome (cardiovascular mortality, myocardial infarction, stroke).** Egger's regression P = 0.039; trim and fill +9 studies. Filled circles represent studies included in the meta-analysis, open circles represent studies added during trim and fill analysis.

**Cardiovascular mortality.** Three cohort studies of sufficient quality were retrieved that compared current users with an untreated or previously treated cohort. All three cohort studies had a retrospective design. Two small studies from the same group (one in patients without gout, and one in patients with gout) observed a reduced cardiovascular and total mortality that was only significant in the study with gout patients [51, 52]. The third study with a much larger sample size compared users with non-users and observed a non-significant reduction in cardiovascular mortality and a significant reduction in total mortality [55].

**Myocardial infarction.** Four cohort studies of sufficient quality were retrieved that compared a cohort of current users with an untreated or previously treated cohort. All four studies have a retrospective study design. In two of these studies, allopurinol treatment was associated with a dose dependent reduction in myocardial infarctions [56, 57]. Another study showed benefit associated with allopurinol treatment, but did not analyse a potential relation with dose [59]. One study did not show any benefit on myocardial infarction, but did so on the combined endpoint of stroke, myocardial infarction and cardiovascular mortality [55]. Unfortunately, the allopurinol dose was not reported in this study.

Four case control studies were retrieved that investigated the relation between use of allopurinol and acute coronary events. All four case-control studies were of good quality. De Abajo et al. observed an adjusted odds ratio of 0.52 (95% CI 0.33 to 0.83) in favour of allopurinol (versus no allopurinol). This benefit was fully driven by men (0.44; 95% CI 0.25 to 0.76) with a lack of benefit in women (0.90; 0.36 to 2.23). The benefit was only observed at higher doses (300 mg or higher) and prolonged treatment duration (> 180 days) [63]. Similar observations have been reported by Rodriguez-Martin et al. regardless the presence of gout [67]. Grimaldi-Bensouda et al. also found a reduced risk in those who used allopurinol (adjusted OR 0.73 (95% CI 0.54 to 0.99) [64]. These investigators did not observe a dose response relationship but the majority (>70%) of the allopurinol users in this study used

**Table 1. Overview of included observational studies.**

| Article ID | Design | Follow-up (weeks) | Number of patients | Dose (mg/day) | Outcomes and adjusted HR (cohort studies)/ OR (case-control) [95% CI] | Comments |
|---|---|---|---|---|---|---|
| Chen 2015a [51] | RCH | 333 | 546 | NR | CM: 0.49 [0.12, 2.00] | All-cause mortality: 1.00 [0.51, 1.95]. Only patients with hyperuricemia without gout. |
| Chen 2015b [52] | RCH | 338 | 572 | NR | CM: 0.37 [0.01, 0.48] | All-cause mortality: 0.39 [0.22, 0.70]. Only patients with hyperuricemia and gout. |
| de Abajo 2015 [63] | CC | NA | 21696 | NR | MI: OR 0.52 [0.33, 0.83] | Sex |
|  |  |  |  |  |  | Men: 0.44 [0.25, 0.76] |
|  |  |  |  |  |  | Women: 0.90 [0.36, 0.88]. |
|  |  |  |  |  |  | Dose |
|  |  |  |  |  |  | ≥300mg: 0.30[0.13, 0.72] |
|  |  |  |  |  |  | <300mg: 0.67[0.37, 1.23] |
| Grimaldi-Bensouda 2015 [64] | CC | NA | 7126 | ≥200: 66% | MI: OR 0.80 [0.59, 1.09] | |
|  |  |  |  | ≥300: 19% |  | |
| Ju 2019 [53] | RCH | 99 | 6938 | 100/200/ 300 mg | CM: 0.727 [0.231, 2.294] | All-cause mortality: 0.975 [0.888, 1.070]. Only patients with gout, without MACE before gout diagnosis |
|  |  |  |  |  | MI: 1.145 [0.873, 1.052] |  |
|  |  |  |  |  | S: 0.831 [0.645, 1.070] |  |
| Lai 2019 [65] | CC | NA | 29574 | NR | S: OR 0.992 [0.989, 0.996] | Cumulative duration of allopurinol use: |
|  |  |  |  |  |  | <1 year: 1.12[1.04, 1.21] |
|  |  |  |  |  |  | 1–3 year: 0.97[0.81, 1.16] |
|  |  |  |  |  |  | >3 years: 0.74[0.57, 0.96] |
| Larsen 2016 [55] | RCH | 264 | 14254 | NR | CM: 0.90 [0.78, 1.03] | All-cause mortality: 0.68 [0.62–0.74] |
|  |  |  |  |  | MI: 0.89 [0.73, 1.08] |  |
|  |  |  |  |  | S: 0.89 [0.75, 1.03] |  |
|  |  |  |  |  | CM+MI+S: 0.89 [0.81, 0.97] |  |
| Liao 2019 [66] | CC | NA | 14070 | <200mg: 79% | MI: OR 2.2 [1.7, 2.7] | Dose |
|  |  |  |  | ≥200mg: 21% |  | <200mg: 2.0 [1.5, 2.6] |
|  |  |  |  |  |  | ≥200mg: 2.5 [1.6, 4.0] |
| Lin 2017 [56] | RCH | NR | 2844 | Expressed in DDD (1 DDD = 400 mg /day) | MI: 1.07 [0.86, 1.33] | Exposure-dependent reduction relative to DDD 0–90: DDD 271–360: aHR 0.25 [0.10, 0.61]; DDD > 360: aHR 0,28 [0.12, 0.63] |
| MacIsaac 2016 [57] | RCH | 311 | 4064 | 100: 35.4% | MI: 0.61 [0.43, 0.87] | MI |
|  |  |  |  | 200: 12.8% | S: 0.50 [0.32, 0.80] | <300 mg: 0.87 [0.56, 1.35] |
|  |  |  |  | 300: 51.2% |  | ≥300 mg: 0.38 [0.22, 0.67] |
|  |  |  |  | 600 or higher: 0.59% |  | Stroke |
|  |  |  |  |  |  | <300 mg: 0.66 [0.40, 1.18] |
|  |  |  |  |  |  | ≥300 mg: 0.29 [0.13, 0.62] |
| Rodriguez-Martin 2019 [67] | CC | NA | 23616 | Among current users: <300: 63% | MI: OR 0.84 [0.73, 0.96] | Dose |
|  |  |  |  |  |  | <300mg: 0.90 [0.76, 1.05] |
|  |  |  |  | ≥300: 37% |  | ≥300mg: 0.75 [0.60, 0.93] |
|  |  |  |  |  |  | Treatment duration |
|  |  |  |  |  |  | <180 days: 1.13 [0.91, 1.39] |
|  |  |  |  |  |  | ≥180days: 0.71 [0.60, 0.84] |
| Singh 2016 [60] | RCH | 101 | 26627 (Number of episodes: 28488) | NR | S: 0.91 [0.83, 0.99] | All included patients were allopurinol prescribed at start of observation. Episodes of allopurinol exposure were compared with episodes without. |
|  |  |  |  |  |  | Duration of exposure: |
|  |  |  |  |  |  | <0.5 year: 1.00 [0.88, 1.14] |
|  |  |  |  |  |  | 0.5–2 years: 0.88 [0.78, 0.99] |
|  |  |  |  |  |  | >2 years: 0.79 [0.65, 0.96]. |

*(Continued)*

**Table 1.** (Continued)

| Article ID | Design | Follow-up (weeks) | Number of patients | Dose (mg/day) | Outcomes and adjusted HR (cohort studies)/ OR (case-control) [95% CI] | Comments |
|---|---|---|---|---|---|---|
| Singh 2017b [59] | RCH | NR | 3724768 person years | NR | MI+S: 0.67 [0.53, 0.84] | Current [new] versus previous allopurinol users. Myocardial infarction and stroke combined. |
| | | | | | | Sensitivity analysis did not find an impact of colchicine use. |
| Wei 2011 [61] | RCH | 291 | 2070 | 100 (n = 449) | CM+MI+S: 0.88 [0.73, 1.05] | Within allopurinol users: 0.63 [0.44, 0.91] for high dose versus low dose |
| | | | | 200 (n = 154) | | |
| | | | | ≥300 (n = 432) | | |
| Yen 2020 [62] | RCH | 234 | 14933 person years | NR | S: 0.70 [0.47, 1.03] | All-cause mortality: 0.35 [0.17, 0.75]. Only patients with gout included. |

RCH = retrospective cohort study, CC = case-control study, n = number of participants, NR = not reported, CM = incidence of cardiovascular mortality, MI = incidence of myocardial infarction, S = incidence of stroke, CI = confidence interval, HR = hazard ratio, OR = odds ratio, DDD = defined daily doses.

only a dose of 200 mg/day or less. These authors also investigated the association between colchicine and myocardial infarction and found no colchicine-related protection (adjusted OR 1.17 (0.70 to 1.93)). In contrast, Liao et al. observed an increased risk of myocardial infarction in patients with prescribed allopurinol: adjusted OR 2.2 (95% CI 1.7–2.7), which occurred at all dose levels and increased with dose [66]. Liao et al. studied only patients with an age of at least 65 years.

**Stroke.** *Six* cohort studies of sufficient quality were retrieved that compared a cohort of current users with an untreated or previously treated cohort (Table 1). All studies have a retrospective study design. Four of these studies also reported myocardial infarction, with one study reporting only the combined outcome of myocardial infarction and stroke [59]. Three of these studies reported a significant beneficial association between use of allopurinol and incidence of strokes [57, 59, 60], and three did not [53, 55, 62]. As for myocardial infarction, MacIsaac et al. reported dose dependence: only a dose of 300 mg/day or higher was associated with a lower risk for strokes [57]. One study only reported on stroke outcome and observed benefit associated with allopurinol in the analysis that was restricted to those who had a duration of exposure of at least half a year [60]. Larsen et al. did not observe a significant relation between allopurinol use and stroke as for myocardial infarction, but a significant association with the combined cardiovascular outcome of stroke, myocardial infarction and mortality [55].

One single case-control study was retrieved that studied the relation between allopurinol therapy and ischemic stroke [65]. They studied 14937 cases of first-time ischemic stroke and compared them with an equal number of age and gender matched controls. They included the cumulative duration of allopurinol use in their analysis. They adjusted for age (which was slightly different between controls and cases despite matching) and found an exposure dependent reduction of adjusted odds ratio for ischemic stroke: 1.12 (1.04–1.21) for exposure < 1 year, 0.97 (0.81–1.16) for 1–3 years and 0.74 (0.57–0.96) for more than 3 years of exposure.

**Combined outcome.** Only two retrospective cohort studies with sufficient quality reported this combined outcome [55, 61]. Larsen et al. observed a significant association between allopurinol use and reduced cardiovascular death and event rate. Unfortunately, they did not study the impact of allopurinol dose on this relationship. Wei et al. showed a similar trend in a much smaller group of patients. Interestingly, within the group of allopurinol users they showed a significant association between allopurinol dose and cardiovascular events (see Table 1).

## Grading of the quality of evidence

Grading of quality of the body of evidence is summarized in Table 2. The quality of evidence derived from the randomized controlled trials was classified as low (stroke and cardiovascular mortality as outcomes) to moderate (myocardial infarction and the combined outcome of myocardial infarction, stroke and cardiovascular mortality. Down-grading of trials was performed because of serious risk of bias (all outcomes) and imprecision (mortality and stroke). None of the trials was specifically designed to detect these outcomes. Twenty out of 26 trials reported zero events. In these 20 trials 1059 patients were treated with allopurinol and 907 patients received placebo. Thus, studies that reported events included only a minority of randomized patients (938 out of the 2904 randomized patients). This is further supported by the funnel plot analysis indicating overrepresentation of studies with a relative risk around 1. The results on stroke and cardiovascular mortality were further downgraded because of the low number of events, resulting in lack of statistical power and therefore imprecision of the results. We did not find significant inconsistency in the results of the included trials. All studies reporting myocardial infarctions showed benefit for allopurinol and no heterogeneity was found. For the combined outcome all but one study reporting events showed benefit for allopurinol, again tests for heterogeneity were insignificant.

## Discussion

This systematic review explored the scientific literature on the impact of allopurinol on cardiovascular outcomes (cardiovascular mortality, myocardial infarction, stroke) in patients with hyperuricemia (with or without gout) and preserved renal function (average MDRD > 30 ml/min/1.73 m$^2$). Both randomized clinical trials (allopurinol versus placebo or no treatment)

**Table 2. Overview of quality assessment of evidence from RCTs according to GRADE-PRO system.**

| Certainty assessment | | | | | | | № of patients | | Effect | | Certainty |
|---|---|---|---|---|---|---|---|---|---|---|---|
| № of studies | Study design | Risk of bias | Inconsistency | Indirectness | Imprecision | Other considerations | allopurinol | no treatment or placebo | Relative (95% CI) | Absolute (95% CI) | |
| 26 | randomised trials | serious [a] | not serious [b] | not serious | not serious | none | 9/1550 (0.6%) | 33/1354 (2.4%) | **RR 0.468** (0.274 to 0.800) | **13 fewer per 1.000** (from 18 fewer to 5 fewer) | ⊕⊕⊕◯; MODERATE |
| 26 | randomised trials | serious [a] | not serious [b] | not serious | serious [c] | none | 10/1550 (0.6%) | 9/1534 (0.6%) | **RR 1.002** (0.557 to 1.805) | **0 fewer per 1.000** (from 3 fewer to 5 more) | ⊕⊕◯◯ LOW |
| 26 | randomised trials | serious [a] | not serious [b] | not serious | serious [c] | none | 20/1550 (1.3%) | 22/1354 (1.6%) | **RR 0.919** (0.560 to 1.508) | **1 fewer per 1.000** (from 7 fewer to 8 more) | ⊕⊕◯◯ LOW |
| 26 | randomised trials | serious [a] | not serious [b] | not serious | not serious | none | 39/1550 (2.5%) | 64/1354 (4.7%) | **RR 0.646** (0.459 to 0.909) | **17 fewer per 1.000** (from 26 fewer to 4 fewer) | ⊕⊕⊕◯ MODERATE |

CI: Confidence interval; RR: Risk ratio; MD: Mean difference; [a] Large number of studies did not have cardiovascular events as a primary outcome measure and the lack of events in many studies without events could possibly be explained by reporter bias; [b] all studies with events showed benefit and no significant heterogeneity was found; [c] we performed a power calculation which suggested at least 2000 patients per treatment arm would have to be included to find a 25% reduction in events. Number of stroke events in the included studies was very low.

and observational studies (retrospective and prospective cohort studies and case-control studies) were included in this review.

## Principal findings

The meta-analysis of available randomized clinical trials showed a significant reduction in the combined outcome (cardiovascular death, myocardial infarction and stroke) that was driven by a significant reduction in myocardial infarction. No significant reduction in cardiovascular mortality or stroke was found. Data retrieved from observational studies was generally consistent with the results from the meta-analysis: the majority of these studies showed that use of allopurinol is associated with reduced cardiovascular events. In these studies, both myocardial infarction and stroke were reduced in patients on allopurinol therapy. Cardiovascular benefit was only observed when allopurinol therapy was prolonged for more than 6 months and when an appropriate allopurinol dose was administered (300 mg or more/day) or sufficient reduction of serum urate concentration was achieved (<0.36 mmol/l).

## Strengths and weaknesses of the study

The quality of the body of evidence retrieved RCTs was low to moderate. Major reasons for down grading were reporting bias and imprecision due to low event-rates. Most studies did not report cardiovascular events as their primary outcome and events had to be extracted from adverse event reporting. Quality of this reporting was often unclear with studies not specifically listing cardiovascular events in their adverse event reporting or not specifying the nature of the event. The observational study data provides some additional insights into limitations of the current meta-analysis. First, the average follow-up time of the trials included in this meta-analysis was around 30–40 weeks, and in many studies < 8 weeks. This may account for the low incidence of cardiovascular events in the retrieved RCTs. Furthermore, two of the six trials with cardiovascular events used a dose less than 300 mg/day [40, 43], while in a third relatively large trial only 69% of the treated patients used the highest applied dose of 300 mg. Interestingly, serum urate concentration was not allowed as a criterium to up-titrate the dose to its maximum allowed dose of 300 mg/day in this trial [44]. This suggests that the meta-analysis may have underestimated the maximum benefit of allopurinol by using suboptimal dose. Although the reviewed observational studies used sophisticated epidemiological techniques to prevent bias as much as possible (propensity score matching to reduce bias by indication and covariate analyses to adjust for residual imbalance in baseline cardiovascular risk between treated and untreated patients), residual bias cannot be excluded. In particular the so called 'healthy adherence bias should be mentioned. Adherence to allopurinol (and therefore dose and therapy duration) could be associated with a healthy lifestyle in general (healthy diet, physical activity, non-smoking). This healthy lifestyle rather than allopurinol exposure could play a role in the observed cardiovascular benefit. Recently, allopurinol has been associated with a slightly increased mortality during the first 30 days after initiation of this therapy [68]. This phenomenon, if confirmed, could have resulted in survival bias, overestimating the benefit of allopurinol adherence.

## Comparison with other studies

Our result differs slightly from a recent meta-analysis by Bredemeier et al. [69]. They did not show a significant impact of allopurinol on major adverse cardiovascular events (MACE) in their overall analysis. However, they observed reduced MACE in the subset of trials that included patients with previous ischemic events. Two important differences between the two studies probably explain this difference: Bredemeier et al. included trials in patients without

hyperuricemia and their meta-analysis did not include the trial recently published by Badve et al. in patients with a high cardiovascular risk profile [44]. In addition, their results suggest that allopurinol doses >300 mg/day may increase cardiovascular events, conflicting with observational study data. As mentioned, they included studies with patients with normal uric acid, and the achieved urate concentration in these patients on allopurinol may very well be much lower than in patients with hyperuricemia that use allopurinol. Experimental studies in humans indicate that extremely low urate concentrations are associated with vascular injury ('J-curve'), hypothetically explained by a direct anti-oxidant effect of uric acid, which could explain these conflicting findings [70].

In addition, 6 other meta-analysis on the comparison between allopurinol and placebo on cardiovascular outcome have been published. Three had a similar approach to ours [71–73]. One reported only all-cause mortality and cardiovascular mortality in 2 observational studies and did not find a consistent effect of allopurinol versus placebo [74]. Two studies had a comparable research question but different patient group: heart failure [75], or patients undergoing cardiac revascularization [76]. Guedes observed a similar effect of allopurinol [71]. They included only one singe RCT and only four observational studies. The RCT is a study with a broad definition of cardiovascular outcome, including hospitalization for heart failure and arrhythmia [22]. This study was replaced in our analysis by a later publication on the same study with a longer follow-up [43]. Ying et al. did not exclude studies in patients without hyperuricemia, and otherwise included less RCT in their database in comparison to our systematic review. These differences may have accounted for their non-significant effect on Major Adverse Cardiovascular Events (MACE). They did not include observational studies [72]. Zhang et al. performed a network meta-analysis including trials that compared allopurinol with febuxostat or placebo [73]. They only included 10 RCTs in their review and did not include some relevant RCTs comparing allopurinol with placebo, nor observational studies. They did not observe a significant effect of either febuxostat nor allopurinol on MACE or cardiovascular mortality.

## Meaning of the study and future research

The present meta-analysis shows a significant reduction in the incidence of combined cardiovascular events in hyperuricemic patients with preserved renal function (eGFR>30 ml/min/1.73 m$^2$) treated with allopurinol. However, there are significant limitations in the quality of the data mainly due to imprecision of results because of its dependence on (likely insufficient) adverse event reporting. Due to these limitations, the results of this meta-analysis do not support its implementation in routine cardiovascular risk management. On the other hand, the analysis provides an important signal for a potentially impressive efficacy of allopurinol to prevent cardiovascular events. Therefore, there is still an urgent need to provide high level trial evidence for the use of allopurinol in cardiovascular protection. Observational study data provide important information on how such a trial should be designed. Based on this systematic review we stress the need for a well-designed randomized double-blind placebo-controlled trial that investigates the benefit of allopurinol on cardiovascular outcome. In this trial, allopurinol should be administered at a dose of 200–400 mg and preferably for at least 3 years. Furthermore, the statistical power should be sufficient to detect a reduction of at least 25% in cardiovascular events.

## Supporting information

**S1 File. Search string.**
(DOCX)

**S1 Table. Characteristics of included studies.** a: RCH = retrospective cohort study, CACO = case-control study, RCT = randomized controlled trial, CROSS = crossover study, NA = not applicable, NR = not reported, CM = incidence of cardiovascular mortality, MI = incidence of myocardial infarction, S = incidence of stroke, ALLO = allopurinol, BENZ = benzbromarone, COL = colchicine; b: ALAT = alanine transaminase, AP = angina pectoris, ASAT = aspartate transaminase, BMI = body mass index, BNP = brain natriuretic peptide, BP = blood pressure, BUN = blood urea nitrogen, CKD = chronic kidney disease, Cr = creatinine, CRP = C-reactive protein, eGFR = estimated glomerular filtration rate, HDL = high-density lipoprotein, HF = heart failure, LDL = low-density lipoprotein, LVEF = left ventricular ejection fraction, NSTEMI = non-ST-elevation myocardial infarction, PAD = peripheral artery disease, STEMI = ST-elevation myocardial infarction, UA = uric acid.
(DOCX)

**S2 Table. Risk of bias CC CH Cross.** a: CACO = case-control study, PCH = prospective cohort study, RCH = retrospective cohort study.
(DOCX)

**S3 Table. Weights of studies in meta-analysis.**
(DOCX)

**S1 Fig. Risk of bias of RCT's.** Green circle represents low risk of bias; yellow circle represents unclear risk of bias; red circle represents high risk of bias.
(DOCX)

**S2 Fig. Forest plots.** a: Overall effect of allopurinol on cardiovascular mortality: p = 0.738; b: Overall effect of allopurinol on myocardial infarction: p = 0.015; c: Overall effect of allopurinol on stroke: p = 0.993.
(DOCX)

**S3 Fig. Funnel plots.** a: Egger's regression P = 0.49; trim and fill +0 studies. Filled circles represent studies included in the meta-analysis, open circles represent studies added during trim and fill analysis; b: Egger's regression P<0.0001; trim and fill +13 studies. Filled circles represent studies included in the meta-analysis, open circles represent studies added during trim and fill analysis; c: Egger's regression P = 0.14; trim and fill +12 studies. Filled circles represent studies included in the meta-analysis, open circles represent studies added during trim and fill analysis.
(DOCX)

**S1 Checklist. PRISMA 2009 checklist.**
(DOC)

## Author Contributions

**Conceptualization:** Kimberley E. Wever, Gerard A. Rongen.

**Data curation:** Karel H. van der Pol, Kimberley E. Wever, Gerard A. Rongen.

**Formal analysis:** Karel H. van der Pol, Kimberley E. Wever, Jan H. Cornel, Gerard A. Rongen.

**Investigation:** Karel H. van der Pol, Kimberley E. Wever, Gerard A. Rongen.

**Methodology:** Kimberley E. Wever, Gerard A. Rongen.

**Project administration:** Karel H. van der Pol, Kimberley E. Wever, Gerard A. Rongen.

**Software:** Karel H. van der Pol.

**Supervision:** Kimberley E. Wever, Jan H. Cornel, Gerard A. Rongen.

**Validation:** Kimberley E. Wever, Jan H. Cornel, Gerard A. Rongen.

**Visualization:** Karel H. van der Pol.

**Writing – original draft:** Karel H. van der Pol, Kimberley E. Wever, Mariette Verbakel, Frank L. J. Visseren, Jan H. Cornel, Gerard A. Rongen.

**Writing – review & editing:** Karel H. van der Pol, Kimberley E. Wever, Mariette Verbakel, Frank L. J. Visseren, Jan H. Cornel, Gerard A. Rongen.

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
