## [Decision Letter · Decision Letter 0]

11 Jun 2021

PONE-D-21-17078

Allopurinol to reduce cardiovascular morbidity and mortality: A systematic review and meta-analysis.

PLOS ONE

Dear Dr. Rongen,

Thank you for submitting your manuscript to PLOS ONE. After careful consideration, we feel that it has merit but does not fully meet PLOS ONE’s publication criteria as it currently stands. Therefore, we invite you to submit a revised version of the manuscript that addresses the points raised during the review process.

We look forward to receiving your revised manuscript.

Kind regards,

Carmine Pizzi

Academic Editor

PLOS ONE

'I have read the journal's policy and the authors of this manuscript have the following competing interests: Dr. Cornel reports grants from ZonMw (LoDoCo 2 trial), personal fees from Amgen (advisory board), personal fees from Servier(advisory board), personal fees from Astra Zeneca(advisory board),  outside the submitted work; All other authors declare: no support from any organisation for the submitted work; no financial relationships with any organisations that might have an interest in the submitted work in the previous three years; no other relationships or activities that could appear to have influenced the submitted work.'

a. Please confirm that this does not alter your adherence to all PLOS ONE policies on sharing data and materials, by including the following statement: "This does not alter our adherence to  PLOS ONE policies on sharing data and materials.” (as detailed online in our guide for authors http://journals.plos.org/plosone/s/competing-interests).  If there are restrictions on sharing of data and/or materials, please state these.

Please note that we cannot proceed with consideration of your article until this information has been declared.

. Review Comments to the Author

Reviewer #1: I’ve read with attention the paper of Rongen et al. that is potentially of interest. The background and aim of the study have been clearly defined. The methodology applied is overall correct, the results are reliable and adequately discussed. My only concern is that during the last 2 years we can find on PubMed a number of meta-analysis reporting the cardiovascular morbidity and mortality associated with allopurinol (plus/minus febuxostat). However, the authors did not explain how their work is different from the previous and contemporanous ones.

Reviewer #2: Interesting and well conducted meta-analysis

Some issues.

Abstract>narrative analysis>this should be detailed

Abstract>I would remove the sentence about quality of evidence in the abstract

Methods>data synthesis strategy. The sentece "For RCTs, all four outcomes were reported in ≥3 trials, therefore meta-analysis was performed using the OpenMeta[analyst] software package" is not clear (therefore ?)

Methods>maybe the authors should pooled together data from RCTs and observational studies adjusted with multivariate analysis (if any) in order to increase sample size.

---

## [Author Response · Author response to Decision Letter 0]

4 Nov 2021

Response to reviewers

Academic editor

1. We have adapted the revised manuscript according to PLOS ONE’s style requirements.

2. We have included the full statement regarding competing interests, and sharing data and materials as you requested in the cover letter reading: 

'I have read the journal's policy and the authors of this manuscript have the following competing interests: Dr. Cornel reports grants from ZonMw (LoDoCo 2 trial), personal fees from Amgen (advisory board), personal fees from Servier (advisory board), personal fees from Astra Zeneca(advisory board), outside the submitted work; All other authors declare: no support from any organisation for the submitted work; no financial relationships with any organisations that might have an interest in the submitted work in the previous three years; no other relationships or activities that could appear to have influenced the submitted work.

This does not alter our adherence to PLOS ONE policies on sharing data and materials.’

3. We have added captions for our Supporting Information files at the end of our manuscript and we have updated any in-text citations accordingly. 

Reviewer #1

We thank the reviewer for his/her positive assessment of our meta-analysis. 

This reviewer expressed a single concern: ‘The authors did not explain how their work is different from the previous and contemporaneous ones’.

Reply:

We have found 14 extra systematic reviews on the effect of Xanthine Oxidase (XO)-inhibitor treatment on cardiovascular morbidity and mortality (1-14).

Six of these are now discussed in the revised manuscript:

Three had a similar approach to ours (11, 12) (5). One reported only all cause mortality and cardiovascular mortality in 2 observational studies and did not find a consistent effect of allopurinol versus placebo (6). Two studies had a comparable research question but different patient group: heart failure (7) or patients undergoing cardiac revascularization (9).

Guedes observed a similar effect of allopurinol (5). They included only one singe RCT and only four observational studies. The RCT is a study with a broad definition of cardiovascular outcome, including hospitalization for heart failure and arrhythmia (15). This study was replaced in our analysis by a later publication on the same study with a longer follow-up (16). 

Ying et al did not exclude studies in patients without hyperuricemia, and otherwise included less RCT in their database in comparison to our systematic review. These differences may have accounted for their non-significant effect on Major Adverse Cardiovascular Events (MACE). They did not include observational studies (11). 

Zhang et al performed a network meta-analysis including trials that compared allopurinol with febuxostat or placebo. They only included 10 RCTs in their review and did not include some relevant RCTs comparing allopurinol with placebo, nor observational studies. They did not observe a significant effect of either febuxostat nor allopurinol on MACE or cardiovascular mortality. 

The other 8 reviews were not relevant and have not been included in the discussion section of our revised manuscript for the following reasons:

Four only reported on comparison between febuxostat and allopurinol (1, 4, 8, 10)

One reported on the effect of allopurinol of cerebral damage after infarction without assessing the risk of ischemic events (2).

Three did not report data on the comparison between allopurinol and placebo (3, 13, 14)

1. Barrientos-Regala M, Macabeo RA, Ramirez-Ragasa R, Pestano NS, Punzalan FER, Tumanan-Mendoza B, et al. The Association of Febuxostat Compared With Allopurinol on Blood Pressure and Major Adverse Cardiac Events Among Adult Patients With Hyperuricemia: A Meta-analysis. J Cardiovasc Pharmacol. 2020;76(4):461-71.

2. Britnell SR, Chillari KA, Brown JN. The Role of Xanthine Oxidase Inhibitors in Patients with History of Stroke: A Systematic Review. Curr Vasc Pharmacol. 2018;16(6):583-8.

3. Cuenca JA, Balda J, Palacio A, Young L, Pillinger MH, Tamariz L. Febuxostat and Cardiovascular Events: A Systematic Review and Meta-Analysis. Int J Rheumatol. 2019;2019:1076189.

4. Gao L, Wang B, Pan Y, Lu Y, Cheng R. Cardiovascular safety of febuxostat compared to allopurinol for the treatment of gout: A systematic and meta-analysis. Clin Cardiol. 2021;44(7):907-16.

5. Guedes M, Esperanca A, Pereira AC, Rego C. What is the effect on cardiovascular events of reducing hyperuricemia with allopurinol? An evidence-based review. Rev Port Cardiol. 2014;33(11):727-32.

6. Hay CA, Prior JA, Belcher J, Mallen CD, Roddy E. Mortality in Patients With Gout Treated With Allopurinol: A Systematic Review and Meta-Analysis. Arthritis Care Res (Hoboken). 2021;73(7):1049-54.

7. Kanbay M, Afsar B, Siriopol D, Dincer N, Erden N, Yilmaz O, et al. Effect of Uric Acid-Lowering Agents on Cardiovascular Outcome in Patients With Heart Failure: A Systematic Review and Meta-Analysis of Clinical Studies. Angiology. 2020;71(4):315-23.

8. Liu CW, Chang WC, Lee CC, Shau WY, Hsu FS, Wang ML, et al. The net clinical benefits of febuxostat versus allopurinol in patients with gout or asymptomatic hyperuricemia - A systematic review and meta-analysis. Nutr Metab Cardiovasc Dis. 2019;29(10):1011-22.

9. Ullah W, Khanal S, Khan R, Basyal B, Munir S, Minalyan A, et al. Efficacy of Allopurinol in Cardiovascular Diseases: A Systematic Review and Meta-Analysis. Cardiol Res. 2020;11(4):226-32.

10. Wang M, Zhang Y, Zhang M, Li H, Wen C, Zhao T, et al. The major cardiovascular events of febuxostat versus allopurinol in treating gout or asymptomatic hyperuricemia: a systematic review and meta-analysis. Ann Palliat Med. 2021.

11. Ying H, Yuan H, Tang X, Guo W, Jiang R, Jiang C. Impact of Serum Uric Acid Lowering and Contemporary Uric Acid-Lowering Therapies on Cardiovascular Outcomes: A Systematic Review and Meta-Analysis. Front Cardiovasc Med. 2021;8:641062.

12. Zhang S, Xu T, Shi Q, Li S, Wang L, An Z, et al. Cardiovascular Safety of Febuxostat and Allopurinol in Hyperuricemic Patients With or Without Gout: A Network Meta-Analysis. Front Med (Lausanne). 2021;8:698437.

13. Zhang T, Pope JE. Cardiovascular effects of urate-lowering therapies in patients with chronic gout: a systematic review and meta-analysis. Rheumatology (Oxford). 2017;56(7):1144-53.

14. Zhao L, Cao L, Zhao TY, Yang X, Zhu XX, Zou HJ, et al. Cardiovascular events in hyperuricemia population and a cardiovascular benefit-risk assessment of urate-lowering therapies: a systematic review and meta-analysis. Chin Med J (Engl). 2020;133(8):982-93.

15. Goicoechea M, de Vinuesa SG, Verdalles U, Ruiz-Caro C, Ampuero J, Rincon A, et al. Effect of allopurinol in chronic kidney disease progression and cardiovascular risk. Clin J Am Soc Nephrol. 2010(8):1388-93.

16. Goicoechea M, de Vinuesa SG, Verdalles U, Verde E, Macias N, Santos A, et al. Allopurinol and Progression of CKD and Cardiovascular Events: Long-term Follow-up of a Randomized Clinical Trial. American Journal of Kidney Diseases. 2015(4):543-9.

Reviewer 2:

We thank this reviewer for his/her compliments about our meta-analysis. The reviewer expressed the following concerns:

1. The narrative analysis should be detailed and the sentence about quality of evidence should be removed.

Reply: we have deleted the sentence on quality of evidence and we have further elaborated on the narrative analysis in the revised version of the abstract.

2. In the methods section, the sentence ‘For RCTs, all four outcomes were reported in at least three trials, therefore meta-analysis was performed using the OpenMeta[analyst] software package’ is not clear (therefore?)

Reply: Thank you for your comment. On re-reading we understand that ‘therefore’ is not logical in this context. We have changed this sentence to the following:

Each of the four outcomes was reported in at least 3 of the retrieved RCTs. In accordance with our predefined data-analysis plan, a meta-analysis was performed on each of these outcomes. The meta-analysis was performed using the OpenMeta[analyst] software package.

3. May be the authors should pool together data from RCTs and observational studies in order to increase sample size.

We have discussed this point again in considerable detail. Our conclusion is that pooling these data neglect the different sources of uncertainty that arise from RCT’s and observational studies. The Funnel-plot of retrieved RCTs clearly demonstrates insufficient report of cardiovascular adverse events in these trials whereas the observational studies are sensitive to residual (unknown) confounding. In our view, a multivariate analysis will not correct for this problem. Because of these concerns we decided not to pool RCT data with observational data.

---

## [Decision Letter · Decision Letter 1]

18 Nov 2021

Allopurinol to reduce cardiovascular morbidity and mortality: A systematic review and meta-analysis.

PONE-D-21-17078R1

Dear Dr. Rongen,

We’re pleased to inform you that your manuscript has been judged scientifically suitable for publication and will be formally accepted for publication once it meets all outstanding technical requirements.

Kind regards,

Carmine Pizzi

Academic Editor

PLOS ONE

---

## [Editor Report · Acceptance letter]

22 Nov 2021

PONE-D-21-17078R1 

Allopurinol to reduce cardiovascular morbidity and mortality: A systematic review and meta-analysis. 

Dear Dr. Rongen:

I'm pleased to inform you that your manuscript has been deemed suitable for publication in PLOS ONE. Congratulations! Your manuscript is now with our production department. 

Kind regards, 

on behalf of

Prof Carmine Pizzi 

Academic Editor

PLOS ONE